# Strong bottom currents in large, deep Lake Geneva generated by higher vertical-mode Poincaré waves
Rafael Sebastian Reiss [1,2] ✉, Ulrich Lemmin[1], Claire Monin [1,3] & David Andrew Barry [1] ✉

Although internal seiches are ubiquitous in large, deep lakes, little is known about the effect of higher vertical-mode seiches on deepwater dynamics. Here, by combining entire summer season current and temperature observations and 3D numerical modeling, we demonstrate that previously undetected vertical mode-two and mode-three Poincaré waves in 309-meter deep Lake Geneva (Switzerland/France) generate bottom-boundary layer currents up to 4 cm s$^{-1}$. Poincaré wave amphidromic patterns revealed three strong cells excited simultaneously. Weak hypolimnetic stratification ($N^2 \approx 10^{-6}$ $s^{-2}$), typical of deep lakes, significantly modified the wave structure by shifting the lower vertical node in the lake's center from ~75-meter depth (without stratification) to ~150-meter depth (with stratification). This shift induces shear in the middle of the hypolimnion and strengthens bottom currents, with important implications for hypolimnetic mixing and sediment-water exchange. Our findings demonstrate that classical concepts based on constant temperature layers cannot correctly characterize higher vertical-mode Poincaré seiches in deep lakes.

Standing internal (gravity) waves, also called internal seiches, are ubiquitous in stratified lakes. They generate shear-driven thermocline mixing[1–3], enhance mixing in the bottom boundary layer[2,4,5], and modulate sediment-water exchange[6–8]. Mortimer[9] introduced a conceptual model for studying basin-scale internal seiche modes in lakes by approximating water column stratification as distinct constant-temperature (density) layers to distinguish different vertical seiche modes. Depending on stratification and basin morphology, different vertical seiche modes (Vn) can occur in lakes, where n nodes separate layers with currents (and isotherms) oscillating in- or out-of-phase[9,10]. The most often observed vertical mode-one (V1) seiche is characterized by two layers, typically the epilimnion and hypolimnion, with opposing currents[11]. Vertical mode-two (V2) seiches have a three-layer structure due to: (i) a wide thermocline[9,12–14] or (ii) chemical gradients producing a chemocline below the thermocline[15]. Reports of higher vertical modes (n > 2) are rare[16]. Comprehensive treatises on the theory of internal gravity waves are provided by Hutter et al.[17] and Sutherland[18].

In large lakes, where the Coriolis force can modify the internal wave field, Coriolis force effects become important when Burger number $S = c/(Lf) < 1$, where f denotes the latitude-dependent Coriolis parameter, c the non-rotating phase speed and L a characteristic length scale. In large lakes, due to Coriolis force, longitudinal seiches transform into shore-hugging Kelvin waves. Transversal seiches become super-inertial Poincaré waves with horizontal water motions describing clockwise-rotating (Northern Hemisphere) ellipses in the lake interior[19] which converge to near-inertial circles as the Burger number approaches zero. Like non-rotational seiches, "standing" Kelvin/Poincaré waves are formed by the superposition of two oppositely-propagating Poincaré/Kelvin waves and rotate around the respective amphidromic points. Therefore, the term "quasi-standing" is often used in this context (for details, see Hutter et al.[17]). In many large lakes, V1 Kelvin and Poincaré waves have been documented (for example, ref.[20–26]). Reports of higher vertical-mode Kelvin or Poincaré waves, on the other hand, are scarce[5,22,27–30]. It has been suggested that this could be due to a lack of adequate observations rather than the nonexistence of these wave modes[8,17].

In deep large lakes, internal seiches drive hypolimnetic and near-bottom mixing[5,31,32]. However, few studies have been carried out in a lake as deep as Lake Geneva (309-m depth; Switzerland/France; Fig. 1a). In the mid-latitude climate belt, where Lake Geneva and many other deep lakes are located, lake stratification during summer still has a noticeable gradient down to 100-m depth. Below that depth, the stratification gradient becomes much smaller and is often ignored. In the present study, lakes that have more than 100-m depth are considered deep. It appears

[1]Ecological Engineering Laboratory (ECOL), Institute of Environmental Engineering (IIE), Faculty of Architecture, Civil and Environmental Engineering (ENAC), Ecole Polytechnique Fédérale de Lausanne (EPFL), 1015 Lausanne, Switzerland. [2]Present address: Department of Earth Sciences, University of Cambridge, Cambridge, CB2 3EQ, UK. [3]Present address: Research Laboratory in Hydrodynamics, Energetics and Atmospheric Environment (LHEEA), Ecole Centrale de Nantes, UMR CNRS 6598, 44321 Nantes, France. ✉e-mail: rr704@cam.ac.uk; andrew.barry@epfl.ch

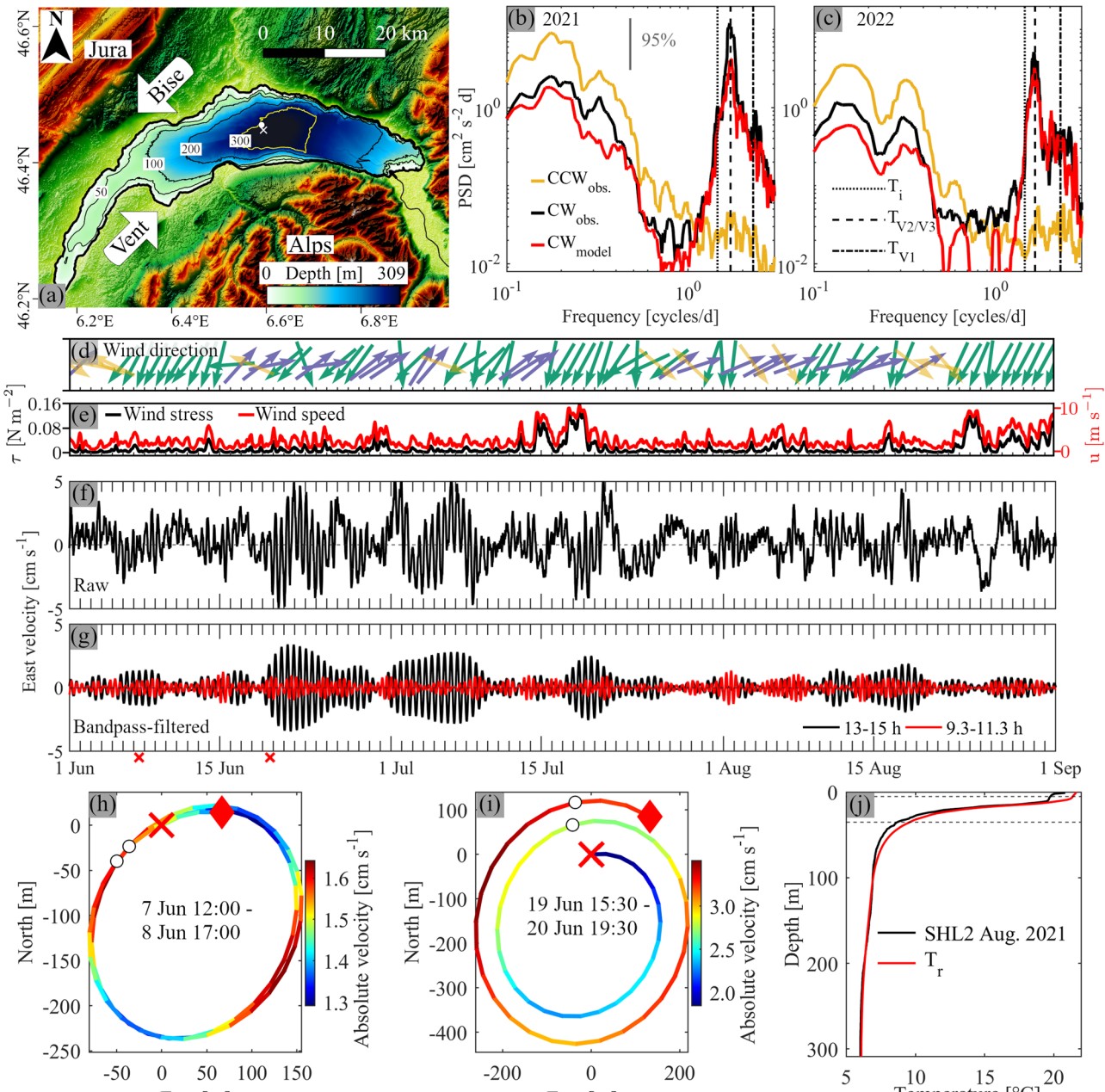

**Fig. 1 | Study site, wind, and field observations of higher vertical-mode Poincaré waves. a** Topographic and bathymetric map of the Lake Geneva area. *Bise* and *Vent*: The two dominant winds that blow over most of the lake surface. White dot: mooring location. White cross: CIPEL monitoring station SHL2[51,73]. **b, c** Measured (black, yellow) and modeled (realistic simulations; red) rotary current spectra at the mooring location (depth-averaged over the lowest ~20 m) from 1 June to 1 September 2021 and 2022, respectively (CCW: counterclockwise; CW: clockwise). Black dashed line: V2/V3 Poincaré period of 14 h (**b**) and 14.5 h (**c**). Black dotted line: Inertial period ($T_i$ = 16.5 h). Black dash-dotted line: V1 Poincaré period ($T_{V1}$ = 10.5 h[23]). The 95% confidence interval is given in **b**. **d** Wind direction and (**e**) wind stress (black) and wind speed (red) from the MeteoSwiss COSMO-1 model averaged over the main basin. For clarity, wind vectors are colored by direction. Green: *Bise* wind; purple: *Vent* wind; yellow: cross-basin winds. **f, g** Measured raw and bandpass-filtered east velocities, depth-averaged over the lowest ~20 m. Black: bandpass-filtered 13–15 h; red: bandpass-filtered 9.3–11.3 h. **h, i** Bandpass-filtered (13–15 h) progressive vector diagrams around 7 and 19 June, respectively; indicated by red crosses in **g**. Black circles are marked every 14 h. Red crosses in **g–i**: start of the progressive vector diagrams; red diamonds: end. **j** Temperature profile measured by CIPEL (black) near the mooring location in August 2021. For comparison, the "realistic" temperature profile, $T_r$, used in the numerical modeling is also shown (red). Horizontal black dashed lines: limits of the thermocline layer. Dates in **d–j** refer to 2021. Close-ups of **b, c** around the near-inertial band are given in Supplementary Fig. 1.

that field studies that have focused on deep hypolimnion hydrodynamics in such deep lakes are rare[8,33]. Therefore, the contribution of internal seiches to deep hypolimnion hydrodynamics of deep lakes remains poorly quantified.

The deep hypolimnion in such large, deep lakes is an important part of the lake system. In Lake Geneva, for example, the volume below 100-m depth is ~48% of the total lake volume and the sediment surface below 100-m depth is ~60% of the total sediment surface. Thus, processes occurring in the deep hypolimnion significantly contribute to the biogeochemical development of the whole lake system. Unlike Kelvin waves that act in the nearshore zone, Poincaré waves in the lake interior can contribute to these deepwater dynamics.

In Lake Michigan (maximum depth 281 m; USA), strong V1 Poincaré wave activity was observed at a 150-m deep midlake location during the entire stratification period[34]. The strongest currents associated with these waves occurred near the surface and could be linked to enhanced lateral dispersion in these layers[35,36]. In contrast, the bottom boundary layers were only weakly affected by V1 Poincaré waves, and no evidence of higher vertical modes was found (see also Ahmed et al.[21]).

In Lake Constance (maximum depth 250 m; Austria, Germany, Switzerland), higher mode longitudinal seiches with wave periods >100 h were studied in a shallow side basin (depth ~100 m) under strong wind forcing[12,37], and V2 Poincaré waves were documented in the upper 60 m in shallow lateral zones[27]. No V2 Poincaré waves in the deep hypolimnion layers were reported. In Lake Iseo (256-m depth; Italy), V2 Poincaré waves were observed, but only investigated in the thermocline and the upper hypolimnion based on temperature measurements[38]. In their analysis of current measurements in the bottom boundary layer in the deepest part of Lake Iseo, Simoncelli et al.[5] found that V1 Poincaré waves contributed to bottom boundary mixing in the deep hypolimnion. In the deepest layers of Lake Garda (maximum depth 346 m; Italy), high rates of turbulence dissipation have been associated with turbulent convection under propagating high-frequency internal waves, which can be triggered by basin-scale internal seiches[39,40]. In Lake Geneva, Lemmin[41] documented ever-present inertial currents near the lake's deepest point (~300-m depth), which contribute to hypolimnetic mixing and sediment-water exchange. Note, however, to our knowledge, vertical mode-three (V3) Poincaré waves have not yet been reported in any of the aforementioned lakes or in any other large, deep lake.

Full-depth current and temperature profiles that allow investigation of details of higher mode Poincaré wave dynamics in the deepest layers of large deep lakes over a full season have thus far not been documented in the literature. In the present study, we therefore combine temperature and current mooring observations covering the full depth range near the deepest point (~300-m depth) in Lake Geneva over two summers with detailed 3D numerical modeling and show that V2 and V3 Poincaré waves are frequently excited over many consecutive wave cycles in the deep hypolimnion. We are then able to explain how these waves dominate the lake's deepwater dynamics during summer and generate currents of up to 4 cm s$^{-1}$ at ~300-m depth. In the deep hypolimnion, the integrated kinetic energy of the V2 and V3 Poincaré wave modes is significantly (~2–2.5 times) higher than that of the V1 Poincaré mode.

Using 3D numerical simulations, we reveal that weak hypolimnetic stratification ($N^2 \approx 10^{-6}\ s^{-2}$; $N$ is the buoyancy frequency), typical of deep lakes, but often ignored, fundamentally changes the vertical structure of higher vertical-mode seiches, inducing shear in the middle of the hypolimnion and strengthening near-bottom currents. We demonstrate that after a short wind impulse of only 4 h duration, V2 and V3 Poincaré waves can persist for more than one week, with integrated kinetic energy progressively shifting towards the inertial period, which explains previously observed near-inertial currents in the deepest layers[41]. Our findings provide ample evidence that higher vertical-mode Poincaré waves make an important contribution to deep-water hydrodynamics, and that they are highly likely to be more widespread in large, deep lakes than the current sparse literature suggests.

## Results and discussion
### Field evidence of Poincaré waves
The current and temperature data collected at a midlake mooring (Fig. 1a) covered two entire stratified summer seasons in 2021 and 2022 (see Methods section). A strong thermocline developed between ~7 and ~35-m depth (Fig. 1j). Rotary spectra of the measured near-bottom currents in the lake's center have a prominent, broad peak in the clockwise-rotating component, which is centered at ~14 h (2021; Fig. 1b) and ~14.5 h (2022; Fig. 1c), and is near the inertial period (~16.5 h). In that frequency range, the spectral power of the counterclockwise rotating component is relatively low compared to the clockwise component, indicating that water mass movement is clockwise. Progressive vector diagrams confirm that horizontal

deepwater motions in this frequency range describe clockwise-rotating ellipses (Fig. 1h, i and Supplementary Fig. 2d, e), thus suggesting that this is an internal wave motion affected by the Coriolis force. However, wave modes within the ~14–15 h period range have not been previously detected in Lake Geneva, raising the question of what causes this dominant current signal.

Bandpass-filtered (13–15 h) velocities show bursts of consecutive oscillations, up to 4 cm s$^{-1}$, in the deepest layers that occur throughout the observation period (Fig. 1g). Combined with background currents, the total near-bottom velocity frequently exceeds 5 cm s$^{-1}$ in short bursts (Fig. 1f). There is no apparent link between current strength and wind speed or direction (Fig. 1d, e, g). The role of the wind is addressed below in the section, *Role of wind forcing*. Oscillating velocities within the ~14–15 h period range are seen throughout the water column, for example, in late June and early July 2022 (Fig. 2f, g). At certain instances, a few days apart, profiles of the horizontal velocities reveal a three- (Fig. 2a) or four-layer (Fig. 2b) current structure. The three-layer current structure resembles a vertical mode-two (V2) internal seiche with two nodes at ~8 and 150–200 m depth which separate layers of opposing currents that reverse direction every half period (compare curves in Fig. 2a, f, g). The four-layer current structure, on the other hand, resembles a vertical mode-three (V3) internal seiche with three nodes at ~13, 25 and 150–200 m depth (see phase differences between adjacent layers in Fig. 2b, f, g).

Bandpass-filtered hypolimnetic temperature measurements show regular periodic isotherm upwelling and downwelling with the same ~14–15 h period occurring continuously and are in phase with the current pattern (Fig. 3). Spectra of temperature have the strongest peak within the ~14–15 h period range coinciding with the peak in current velocities (see Supplementary Fig. 2c). The observed three- and four-layer current structures, the matching temperature and current fluctuations, and the clockwise rotating currents are features that suggest that the dominant ~14–15 h signals measured near the lake's bottom are caused by V2 and V3 Poincaré waves.

Using an analytically-based dispersion relation[33] for an elliptic basin similar to Lake Geneva's deep main basin (Fig. 1a), V2 and V3 Poincaré wave periods of ~14.7 h and 15.7 h, respectively, are obtained for the mean summer 2022 stratification (Supplementary Text 1) in good agreement with our observations, corroborating that these signals are likely due to V2 and V3 Poincaré waves.

Peaks in the clockwise rotating component at periods of ~10–11 h in the measured and modeled spectra (Fig. 1b, c) indicate the presence of V1 Poincaré waves that were previously observed in Lake Geneva[23,42]. However, V1 Poincaré spectral peaks are an order of magnitude smaller than the peak in the ~14–15 h period range. The two-layer current structure of V1 Poincaré waves is confirmed by full-depth current measurements (Fig. 2c). Comparing the observed V1 and V2/V3 wave current profiles, it can be seen that in the upper layer, the velocities of these modes are of the same order of magnitude, as reported in the literature[43]. However, in the deep hypolimnion, close to the bed, V2 and V3 currents are significantly stronger than V1 currents.

This major difference between V1 and V2/V3 current amplitudes is even more pronounced in the full-season time series (Fig. 1g), suggesting that V1 wave-generated currents do not contribute significantly to deep hypolimnion dynamics. The strong increase in V2 and V3 currents when approaching the bed indicates that they can contribute to sediment-water exchange dynamics. The regular and continuous change in V2 and V3 Poincaré current strength and direction (Fig. 1g–i) produces shear that contributes to mixing in the bottom boundary layer. The reason for this V2 and V3 Poincaré wave current increase towards the lakebed is discussed below in the section, *Hypolimnetic stratification impacts vertical wave structure and bottom current strength*. Note that the observed regular, continuous excitation of V2 and V3 Poincaré waves and the dominance of the associated spectral peak over the V1 peak (Fig. 1b, c) are different than those reported in the literature where V2 Poincaré waves are excited after V1 Poincaré waves and occur only in short bursts[43].

More details of the current structure are difficult to determine from these observations because: (i) no measurements are available in the topmost 6 m

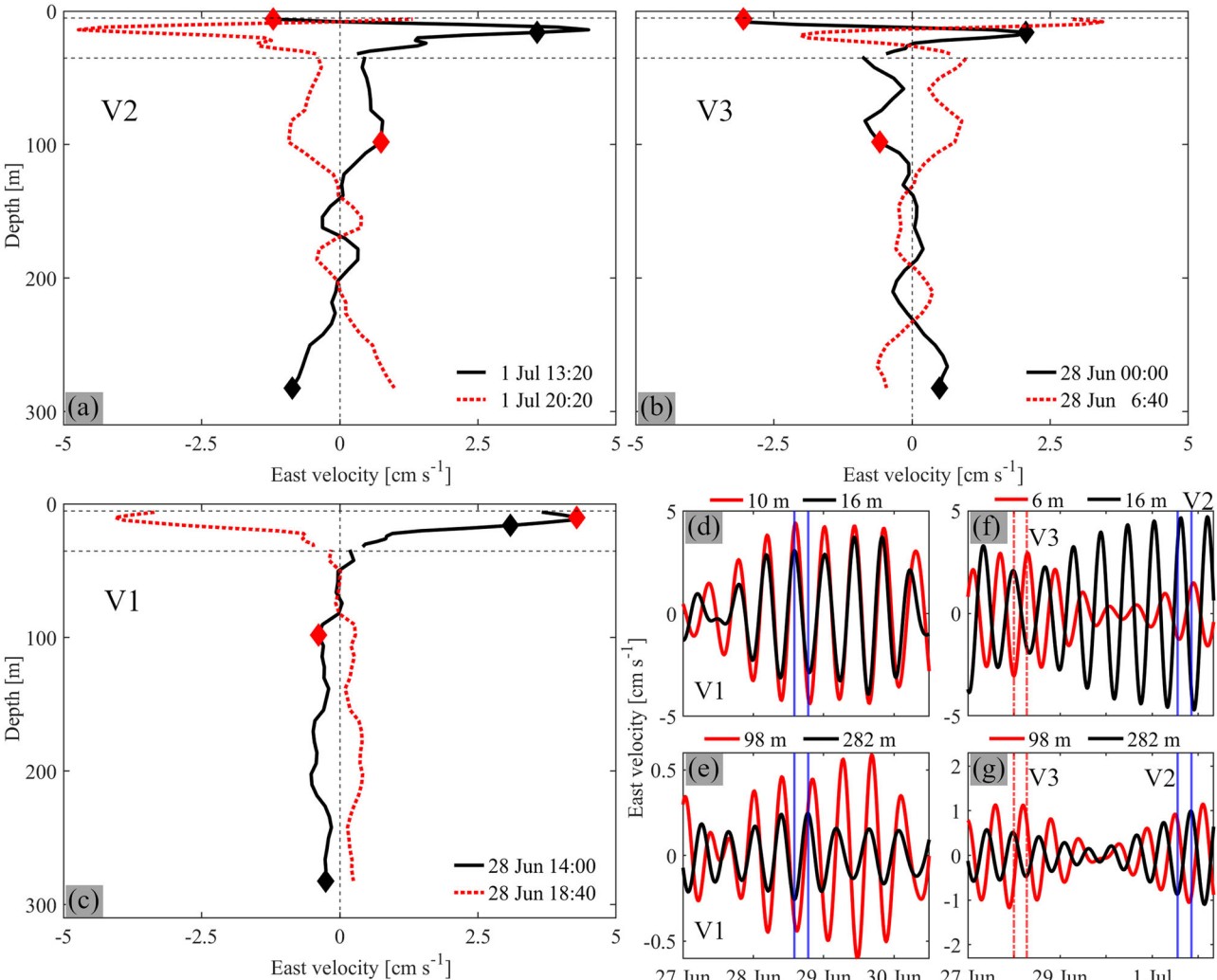

**Fig. 2 | Measured vertical current structure. a–c** Measured east velocity profiles of V1, V2 and V3 Poincaré waves at selected times (see legends), bandpass-filtered between 13.5 and 15.5 h (**a**, **b**; V2, V3), and 9.3 and 11.3 h (**c**; V1). Red and black diamonds: depths of the curves in **d–g**. The thermocline is located between the horizontal black dashed lines. **d–g** Corresponding measured time series at selected depths (see legend), bandpass-filtered between 9.3–11.3 h (**d**, **e**) and 13.5–15.5 h (**f**, **g**). Vertical blue lines in **d**, **e**: time of profiles in **c**. Vertical red dash-dotted and blue lines in **f**, **g**: time of profiles in **b** and **a**, respectively. Measurements were taken at the mooring location in 2022 (for location, see Fig. 1a).

due to surface reflections of the Acoustic Doppler Current Profiler signal, and (ii) low acoustic backscatter at mid-depth (insufficient particles in the water column) produces a noisy signal between ~100 and 200-m depth. Furthermore, in large lakes, different horizontal Poincaré modes with similar periods are often excited simultaneously[24]. Since these different horizontal Poincaré modes cannot be disentangled spectrally due to their similar periods[21,23,44], modal decomposition methods are often employed (for example, refs. 12,29,45–47). Here, using a calibrated 3D hydrodynamic model[42], we carried out realistic simulations (see Methods) that reproduced the clockwise rotating 14–15 h oscillations during the summers of 2021 and 2022 (Fig. 1b, c). Based on the thus validated model, idealized simulations were performed and are analyzed below for V2 and V3 Poincaré wave features.

### V1, V2 and V3 Poincaré wave features revealed by idealized simulations

The idealized simulations (for details, see Methods) were forced with a *Bise* wind impulse (Fig. 1a; from the northeast) and initialized with the realistic initial temperature profile, $T_r$, which was obtained by averaging the model results of the 2022 realistic simulation near station SHL2 (Fig. 1a). It agrees well with the measured temperature profile (Fig. 1j). All near-bottom currents obtained from the idealized simulations show a strong peak in the

clockwise rotary spectra within the ~14–15 h period range and a smaller peak at ~9.5 h (Fig. 4a), similar to rotary spectra of the measured currents and those produced by realistic simulations (Fig. 1b, c). Due to the short duration of the time series, the limited frequency resolution of the model rotary spectra cannot resolve the two neighboring peaks associated with V2 and V3 Poincaré waves. Instead, the respective wave periods were obtained by manually changing the center period of a 0.5-h narrow bandpass-filter in increments of 0.1 h to maximize the thus filtered near-bottom velocities. The following wave periods were determined: V1: ~9.3 h, V2: ~14.3 h, and V3: ~15.0 h. In the sections below, the bandpass-filtered model results are analyzed with narrow filter bands of 9.05–9.55 h (V1), 14.05–14.55 h (V2), and 14.75–15.25 h (V3). Note that no signals of vertical Poincaré modes higher than V3 were found in the observations or model results, suggesting that they do not play an important role in Lake Geneva.

Horizontal velocity profiles bandpass-filtered around 9.3 h have a two-layer pattern with a current reversal just below the thermocline (Fig. 8a). This pattern, also seen in the field observations (Fig. 2c), is typical of V1 Poincaré modes, as discussed for Lake Geneva by Lemmin et al.[23]

Horizontal velocity profiles bandpass-filtered around 14.3 h have a three-layer structure with two nodes: one within the thermocline at ~15-m depth and one in the middle of the hypolimnion at ~150-m depth (Fig. 4d).

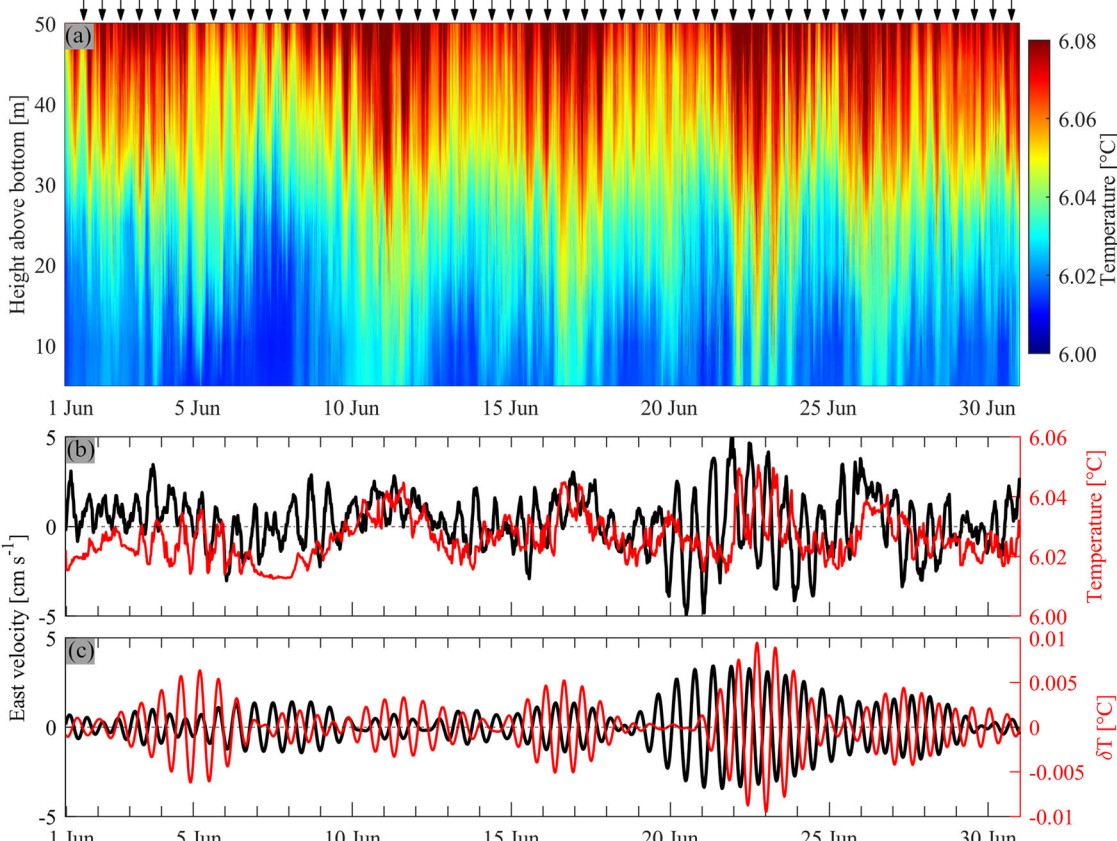

**Fig. 3 | Temperature observations. a** Temperature contours 0–50 m above the lakebed (309-m depth). Black arrows on the top of the panel are given every 14 h. **b**, **c** Raw and bandpass-filtered (13–15 h) east velocities, depth-averaged over the lowest ~20 m (black) and temperatures (temperature variations, $\delta T$, in panel **c**) 15 m above the lakebed (red). Measurements were taken at the mooring location in 2021 (for location, see Fig. 1a).

Moreover, currents in adjacent layers are 180° out-of-phase (Fig. 4b, c). Profiles of vertical isotherm displacements show one node in the center of the thermocline, with isotherms in the epilimnion and upper thermocline moving vertically in opposite directions compared to deeper layers (Fig. 5e). Isotherm excursions are largest in the weakly stratified hypolimnion. Such a three-layer current structure with periodic compression and expansion of the middle layer is characteristic of V2 seiches[14,17,48].

Horizontal velocity profiles bandpass-filtered around 15.0 h, on the other hand, have three nodes at ~10, 25, and 160-m depth (Fig. 4h) that separate four layers with currents in adjacent layers 180° out-of-phase (Fig. 4f, g). Typical of V3 internal seiches, profiles of vertical isotherm displacement show two nodes, one at the center of the thermocline and one at its lower end (Fig. 5f). Here, hypolimnetic waters move vertically in phase with the epilimnion and upper thermocline layers, and out-of-phase with the lower thermocline layers.

The clockwise-rotating currents (Fig. 1h, i) and the multi-layer structures in the profiles of isotherm displacement and horizontal currents (Figs. 5e, f and 4d, h) confirm that the dominant ~14-15 h signals are caused by V2 and V3 Poincaré waves. The observations and model results at the mooring location suggest that modes V2 and V3 are comparably strong, with very similar shapes in the hypolimnion; significant differences only appear above ~50-m depth (Fig. 4d, h). The three-dimensional structure of these wave modes and their relative strength compared to the V1 Poincaré mode will be discussed in the following two sections.

**Horizontal and vertical structure of V2 and V3 Poincaré waves**

In large lakes, Poincaré waves are horizontally characterized by one (horizontal mode-one) or several (higher horizontal modes) circular cells with maximum velocities at the cell centers and maximum isotherm displacements at the boundaries. Each cell rotates around an amphidromic point where vertical velocities and thus isotherm displacements vanish[21,23]. For both modes V2 and V3, vertical isotherm displacement timelines at 35-m depth suggest several large cells in the main basin, with the amphidromic points approximately located along the lake's central axis (Fig. 5a, grey lines). In addition to these large cells, several smaller cells are found near the shores (see, for example, at 20-km East and 18-km North in Fig. 5a). These results show that V2 and V3 Poincaré waves of higher horizontal modes in along- and cross-basin directions (see, e.g., ref. 24) are simultaneously excited by a wind event over most of the main lake basin. As is typical for Poincaré waves, isotherm displacements are strongest near the shores (Supplementary Figs 4, 5).

Similar to the isotherm displacements, several larger cells appear along the lake's central axis in the depth-averaged (V2: 20–25 m, and V3: 15–20 m) horizontal current velocity maps, with currents strongest near the center, diminishing shoreward (Fig. 5a, b). Furthermore, smaller, significantly weaker cells are found near the shores, in agreement with the isotherm displacement patterns. The discussed amphidromic patterns characterized by several larger cells centered along the lake's longitudinal axis and smaller cells near the shores are similar to the amphidromic patterns of V1 waves[23], which were confirmed by field measurements. However, in contrast to V1 Poincaré waves[23], V2 and V3 Poincaré waves are limited to Lake Geneva's deep central basin (Fig. 5a, b).

Currents along a vertical transect through the ~300-m deep mooring location (red line in Fig. 5a, b) confirm the three- (V2) and four-layer (V3 Poincaré wave) structures in the previously discussed velocity profiles (Fig. 4d, h). The depths of the lower nodal lines and thus the thickness of the current layers in the hypolimnion vary significantly along the transect, with the maximum nodal depth of ~150 m found in the deepest regions of the lake (Fig. 5c, d). Similar shoreward shoaling of the lower nodal lines occurs at other transects throughout the lake (e.g., Supplementary Fig. 6c, d).

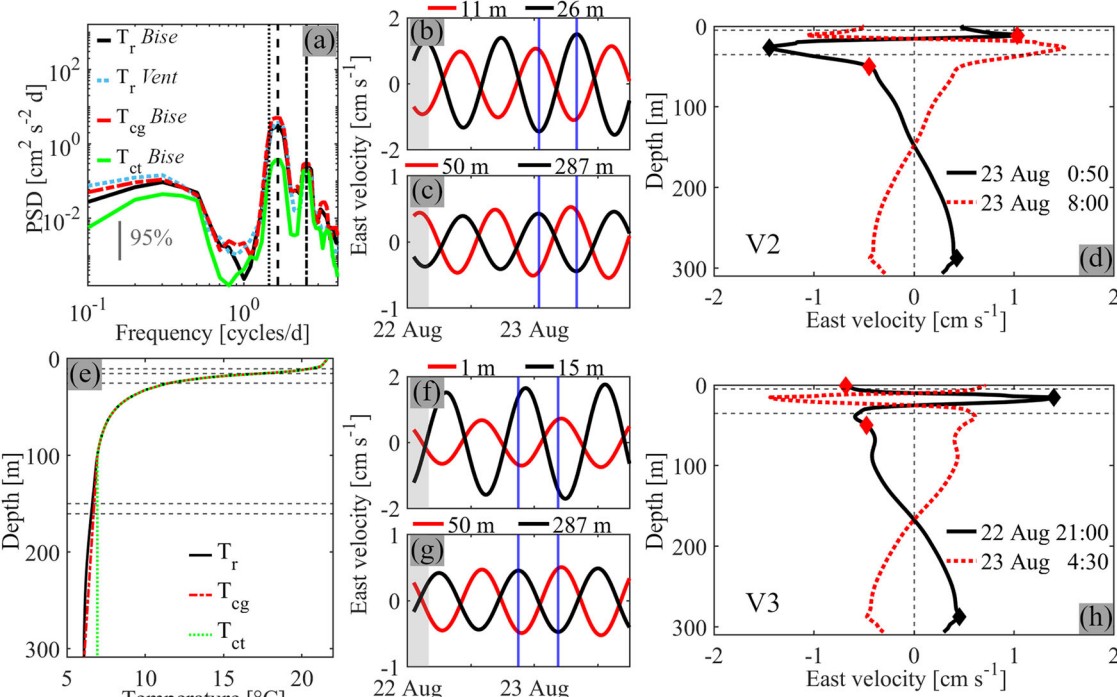

**Fig. 4 | Idealized simulations of V2 and V3 Poincaré waves at the mooring location. a** Clockwise rotary spectra of the near-bottom currents for different initial temperature profiles ($T_X$) and wind conditions (*Bise* or *Vent*). Vertical lines: 16.5 h (inertial period; dotted), 14.5 h (V2/V3 Poincaré; dashed), 9.5 h (V1 Poincaré; dashed-dotted). The 95% confidence interval is given. Compare with spectra in Fig. 1b, c for field measurements and realistic simulations. **b, c** East velocities at selected depths (see legends). Vertical blue lines: time of the profiles in **d**. Grey shaded area: 4-h wind impulse. **d** East velocity profiles at selected times (see legend). Red and black diamonds: depths of the curves in **b, c**. Horizontal dashed lines:

thermocline layer. Modeling results in **b–d** are 14.05–14.55 h bandpass-filtered (V2 Poincaré). **e** Initial temperature profiles. $T_r$: from the realistic simulations. $T_{cg}$: same as $T_r$ but with constant hypolimnetic stratification. $T_{ct}$: same as $T_r$ but with constant hypolimnetic temperatures. Horizontal dashed lines: depths of the vertical nodes in **d, h** (10, 15, 25, 150, 160 m). The corresponding density and buoyancy frequency profiles are given in Supplementary Fig. 3. **f–h** Same as **b–d** but bandpass-filtered between 14.75–15.25 h (V3 Poincaré). Simulations were forced with a 4-h *Bise* wind impulse and initialized with temperature profile $T_r$ (see panel **e**).

---

Furthermore, at shallower transects, the lower nodal line is generally found at shallower depths. For example, in the cell near the lake's center (~300-m depth), the maximum nodal depth is ~150 m, whereas it is only ~75 m at the strong westernmost cell (~150-m depth; Supplementary Fig. 6c, d). The time evolution of the currents at different transects is shown in Supplementary Movie 1.

The vertical current transects demonstrate that the currents induced by V2 and V3 Poincaré waves exhibit a strong spatial heterogeneity, with the vertical current structure in the hypolimnion significantly impacted by sloping topography and local depth. The great depth of the lower nodal lines and the horizontal and vertical heterogeneities produce shear that can contribute to mixing in the deep hypolimnion.

The amphidromic points do not always coincide with the strongest currents (Fig. 5a, b). This mismatch is likely due to the superposition of different horizontal modes that cannot be distinguished spectrally. The dispersion relation for Poincaré waves in a rectangular basin is given as $T_{PW}^{-2} = T_i^{-2} + \left(1 + r^2\right) T_{nr}^{-2}$, where $T_{PW}$ denotes the Poincaré wave period, $T_i$ the inertial period, $r$ the basin aspect ratio, and $T_{nr}$ the nonrotational cross-basin wave period[24,49]. Thus, $T_{PW}$ is typically close to $T_i$, but does not exceed it. For very large lakes, such as the Laurentian Great Lakes (Canada, USA), $T_{PW}$ approaches $T_i$ and different horizontal-mode Poincaré waves have nearly identical periods[21,44,50]. For Lake Geneva, it was demonstrated that different horizontal-mode V1 Poincaré waves have similar periods that cannot be distinguished spectrally[23]. Our results suggest this also holds for higher vertical-mode Poincaré waves. Note that in very large lakes, where all Poincaré modes converge to the inertial frequency, $T_i$, narrow bandpass-filtering, as employed here for Lake Geneva, cannot effectively separate different vertical modes. In that case, the contribution of different vertical modes can be estimated by decomposing the observed or modeled

baroclinic currents into a linear combination of the theoretical vertical modal shapes obtained by solving the Taylor-Goldstein equation[25].

The cell-like isotherm displacement and current patterns, and the three- and four-layer current structures obtained from the idealized simulations, again corroborate that the ~14–15 h signals observed and modeled in the lake's deepest layers during summers 2021 and 2022 are due to previously unknown V2 and V3 Poincaré waves of higher horizontal order. Disentangling the complex horizontal structure of these V2 and V3 Poincaré waves in Lake Geneva is beyond the scope of this study.

### Relative strength of V1, V2, and V3 Poincaré waves

The full-season observations and realistic simulations for summers 2021 and 2022 suggest that the currents produced by Poincaré modes V2 and V3 are significantly stronger in the bottom layers than those of mode V1 (Fig. 1). This is confirmed by the basin-wide integrated kinetic energy associated with the different modes obtained from the bandpass-filtered idealized model results. When considering the entire depth range, V1 Poincaré mode energy is twice as high as that of V2 and V3 modes during the first ~2 d after the wind impulse (Fig. 6a). However, when considering only the layers below 100-m depth, V1 energy is as high as V2 energy only during the first ~1.5 d. Thereafter, V1 mode energy quickly dies down, whereas V2 and V3 mode energies get stronger for 2-3 d, peaking at ~4 d (V2) and ~5 d (V3) after the simulation started (Fig. 6b). This temporal lag between V1 and V2/V3 modes is similar to the observations of Wiegand and Chamberlain[43].

The maximum hypolimnion-integrated kinetic energy of the V2 and V3 mode is ~40% (V2) and 20% (V3) higher than that of the V1 mode. More striking is that the time-integrated hypolimnetic kinetic energy of the higher vertical modes during the first week of the simulation is 2.3 (V2) and 2.1

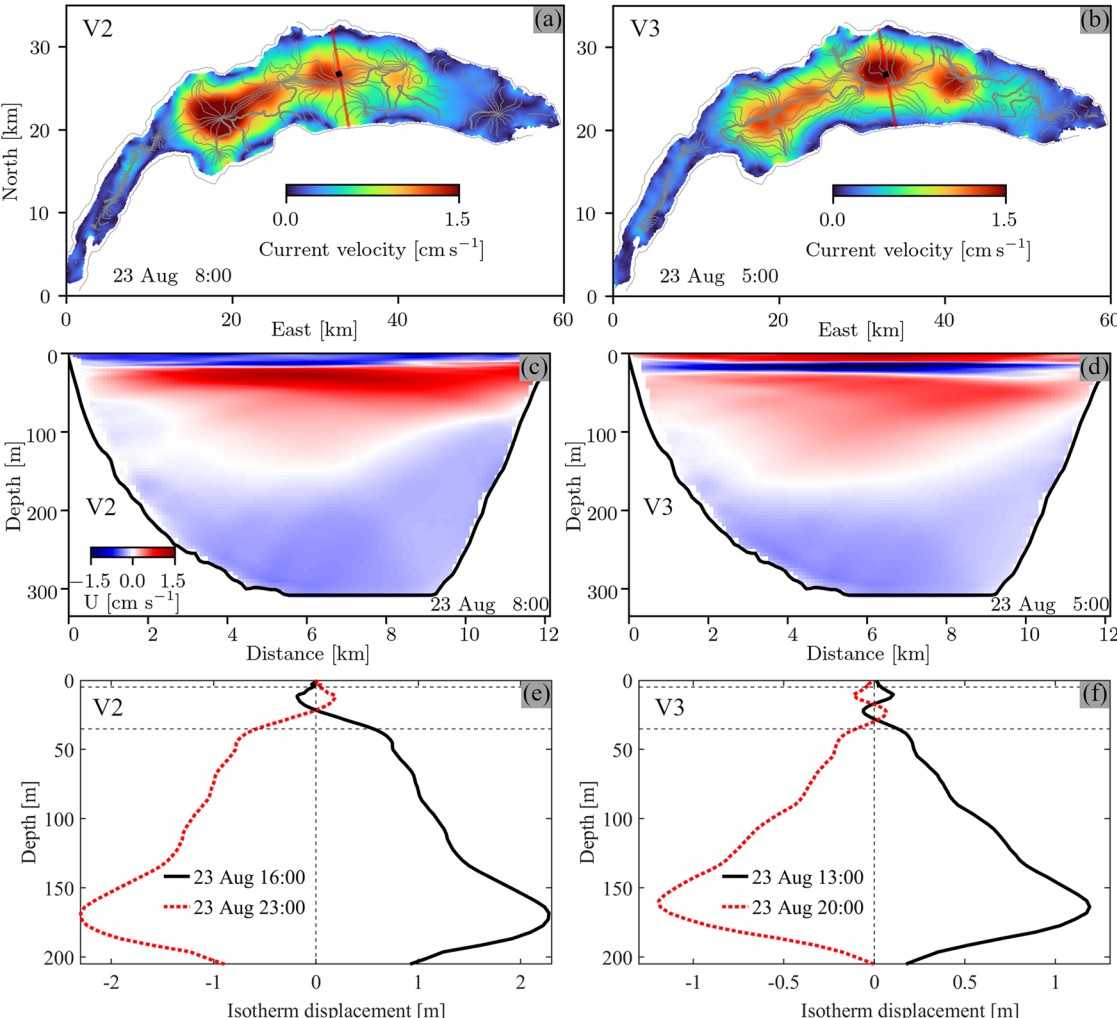

**Fig. 5 | Horizontal and vertical structure of higher vertical-mode Poincaré waves.** Left column: Vertical mode-two (V2). Right column: Vertical mode-three (V3). **a, b** Timelines of vertical isotherm displacement ($d_{iso}$) at ~35-m depth (grey lines) and depth-averaged (V2: 20–25 m, and V3: 15–20 m) current speeds. The $d_{iso}$ timelines show the crest arrival time during one wave period (1-h intervals). Black dot: mooring location. Red line: location of the transversal transect in **c, d**. **c, d** Current velocities into (red) and out of (blue) the plane along the transect shown in **a, b**. Distance is measured from the northern shore. **e, f** Vertical isotherm displacements, relative to the initial temperature profile, $T_r$, at a ~ 200-m depth location north of the mooring location. Horizontal dashed lines: thermocline layer. The model was forced with a 4-h *Bise* wind impulse and initialized with temperature profile $T_r$ (Fig. 4e). Modeling results are bandpass-filtered (V2: 14.05–14.55 h, and V3: 14.75–15.25 h).

(V3) times higher than that of the first vertical mode. This analysis confirms that V2 and V3 Poincaré waves play an important, but as yet overlooked role in the deepwater dynamics of Lake Geneva, whereas V1 Poincaré waves are more relevant in the upper layers. The V3 mode becomes stronger than the V2 mode in the hypolimnion after ~4-5 d, as is reflected in a shift of the dominant oscillation period towards longer periods in the bottom layers during the first week of the simulation (Supplementary Fig. 7). Thus, after an initial short wind impulse, the oscillation period gradually approaches the inertial period, which can explain previously observed near-inertial currents in the deepest layers of Lake Geneva[41].

## Beat pulsation by superposition of V2 and V3 Poincaré waves
The superposition of two oscillations with similar strength and periods can lead to so-called beat pulsation, a well-known phenomenon in acoustics. Both the V2 and V3 Poincaré modes are initiated simultaneously by the 4-h wind impulse (Fig. 6c). However, as the waves evolve, the relative phase difference continuously changes due to the small difference in the wave periods. This causes alternating positive and negative interference between the two modes (see + and − signs in Fig. 6c, d) and manifests itself as a typical beat pattern in the combined signal (Fig. 6d). Mortimer[24] showed

that beat pulsation due to different horizontal-mode Poincaré waves is a common phenomenon in Lake Ontario and Lake Michigan. Our results demonstrate that beat pulsation also occurs due to superposition of different vertical-mode Poincaré waves in Lake Geneva. In addition to time-varying wind forcing, such beat pulsations can be one explanation for the pulsating current patterns seen in the full-season observations (Fig. 1g).

## Hypolimnetic stratification impacts vertical wave structure and bottom current strength
Twice a month monitoring by the *Commission Internationale pour la Protection des Eaux du Léman* (CIPEL[51]) over several decades shows that a weak mean stratification, $N^2$, of $O(10^{-6}\ s^{-2})$[51] is always present during summer in Lake Geneva's deep hypolimnion. This was also confirmed by the present long-term timeseries data where slight depth variations of this mean gradient were observed[41]. To investigate the effect of deep hypolimnion stratification on the vertical structure of V2 and V3 Poincaré waves, idealized simulations were carried out with different stratification profiles in the deep hypolimnion below 100-m depth.

For the simulation initialized with the realistic temperature profile $T_r$, the lower node of the V2 and V3 modes in the lake's ~300-m deep center is

**Fig. 6 | Strength of V1, V2 and V3 Poincaré waves, and their interaction.** Kinetic energy (KE) of the different modes integrated over the entire lake (**a**) over the full depth range and (**b**) below 100-m depth. KE was computed based on the bandpass-filtered east and north velocity components. **c** Near-bottom velocities at the mooring location obtained with narrow bandpass filters (legend in **a**). **d** Same as **c** but filtered with a wider bandpass filter (13.5–15.5 h) that includes both the V2 and V3 wave periods. The magenta vertical lines in **c**, **d** mark times of positive (+) and negative (−) interference between modes V2 and V3, causing beat pulsation (compare panels **c**, **d**). The simulation was forced with a 4-h *Bise* wind impulse (grey shaded area) and initialized with temperature profile $T_r$ (Fig. 4e). Results in **a**–**c** are filtered with a 0.5-h narrow passband (V1: 9.05–9.55 h, V2: 14.05–14.55 h, and V3: 14.75–15.25 h). Idealized simulation for 2022.

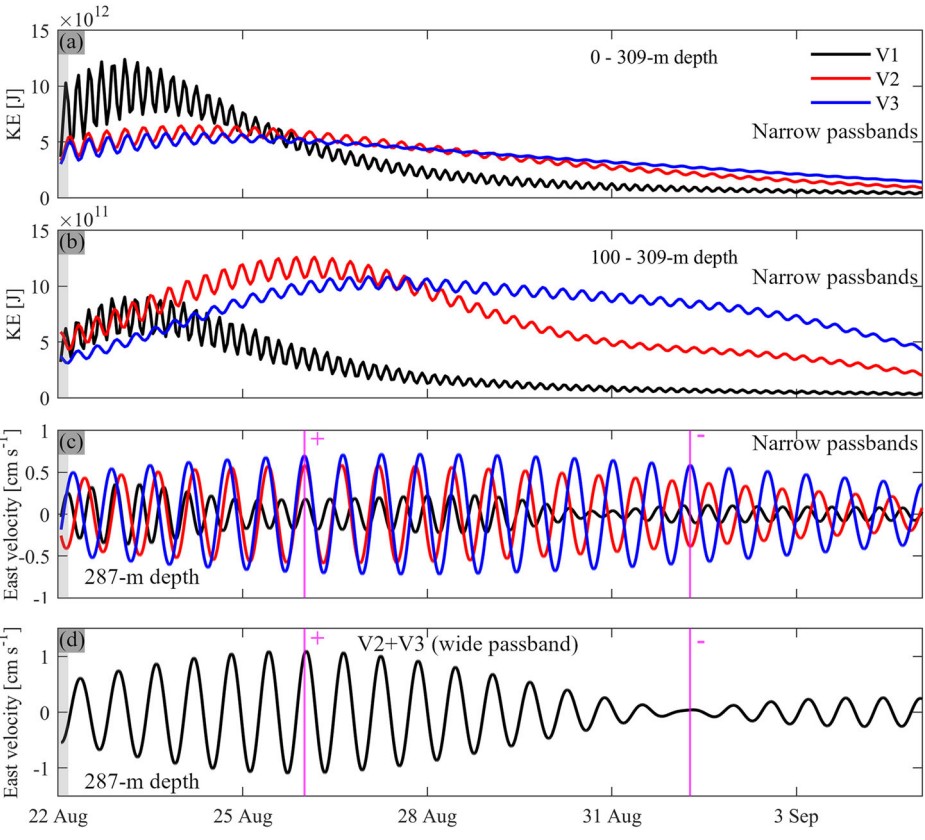

located well within the hypolimnion at a maximum depth of ~150 m (Fig. 7a–d). Below that depth, only a weak temperature gradient exists, which changes slightly at ~200-m depth (see profile $T_r$ in Fig. 4e). When the model is initialized with a temperature profile that is identical to $T_r$ above 100-m depth, but has a constant weak temperature gradient ($N^2 \approx 10^{-6}\,s^{-2}$; $N^2$ profile in Supplementary Fig. 3) below 100-m depth (profile $T_{cg}$ in Fig. 4e), the results for both the V2 and V3 modes are nearly identical to those obtained for the simulation initialized with $T_r$ (velocity profiles in Fig. 7a, b and rotary spectra in Fig. 4a).

However, when hypolimnetic stratification is entirely removed (constant temperatures below 100-m depth; profile $T_{ct}$ in Fig. 4e), the vertical current structure changes significantly. In particular, the lower node in the lake's center, which was located at a maximum depth of ~150 m with stratification, moved up to ~75-m depth (Fig. 7; transects at other locations in Supplementary Figs 6 and 8), that is, to the depth range where the thermocline transitions into the now unstratified hypolimnion. Moreover, without hypolimnetic stratification, the three- (V2) and four-layer (V3) current structures closely followed the stratification, as expected from the literature[52]. However, the resulting lower layer currents were considerably weaker than when weak hypolimnetic stratification was present. Without hypolimnetic stratification, V2 and V3 near-bottom currents at the mooring location weakened by a factor of up to 2.4 (V2) and 3.4 (V3) (Fig. 7a, b; also see rotary spectra in Fig. 4a). Note that without hypolimnetic stratification, no vertical gradients of velocity exist in the hypolimnion, except in the frictional bottom boundary layer (Fig. 7a, b).

The Taylor-Goldstein equation can be used to determine the profile shape of different vertical modes for different stratification profiles[53]. Good agreement is found between the bandpass-filtered profiles of the horizontal velocities from the 3D numerical simulations discussed above (Fig. 8, left column) and the theoretical modal shapes of the V1, V2, and V3 modes obtained by solving the Taylor-Goldstein equation for the three different stratification profiles (Fig. 8, right column). Furthermore, the shapes of the eigenfunctions of the vertical velocity for both V2 and V3 agree well with the profiles of the bandpass-filtered modeled vertical

isotherm displacements (Fig. 5e, f and Supplementary Fig. 9). This confirms that the newly discovered ~14–15 h oscillations in Lake Geneva are V2 and V3 internal seiches and corroborates the important impact that weak hypolimnetic stratification has on the modal shape of higher vertical-mode seiches in deep lakes. Different from V2 and V3 modes, the shape of the V1 mode is not significantly affected by weak hypolimnion stratification (Fig. 8a, b).

Our results make evident that this ever-present weak hypolimnetic stratification, typical for deep lakes, significantly modifies the structure of higher vertical-mode seiches. In Lake Geneva, hypolimnetic stratification caused the greater depth of the lower V2 and V3 Poincaré wave nodes and strengthened near-bottom currents. The deep node within the weakly stratified hypolimnion induces shear in the water column that can contribute to hypolimnetic mixing in the intermediate layers. Furthermore, enhanced near-bottom currents strengthen hypolimnetic mixing in the bottom boundary layer[4,5] and affect sediment-water exchange processes that modulate dissolved oxygen consumption[7,54]. The above findings are particularly important since higher vertical-mode seiches are often studied with simplified models based on multiple layers of constant density[52,55,56]; this in turn, produces unrealistic hypolimnion current patterns.

### Role of wind forcing

Previous observations have reported that internal waves are initiated in the upper layers of the lake after a strong wind impulse[12,27,43]. Changing the 4-h wind impulse from a *Bise* wind (from the northeast; Supplementary Fig. 10a, b, e) to a similarly strong *Vent* wind (from the southwest; Supplementary Fig. 10c, d, f) did not significantly change the V2 and V3 Poincaré wave structure or strength (Fig. 4a). This suggests that wind direction has no effect on their generation, in agreement with the measurements (Fig. 1). However, winds lasting longer than ~8 h (half the inertial period) are less efficient in generating Poincaré waves, as previously suggested[57]. Thus, wind field fluctuations of O(4–8 h) would excite V2 and V3 Poincaré waves more efficiently[23].

**Fig. 7 | Effect of hypolimnetic stratification on vertical wave structure.** Left column: Vertical mode-two (V2). Right column: Vertical mode-three (V3). **a**, **b** East velocity profiles at the mooring location (see Fig. 5a), ~1.5 d after the simulation started with initial temperature profiles $T_r$ (black), $T_{cg}$ (red), and $T_{ct}$ (green) (for profiles, see Fig. 4e). The inset shows the corresponding time series depth-averaged over the lowest ~30 m (vertical blue dashed line: time of the profiles in **a**, **b**). **c–f** Current velocities into (red) and out of (blue) the plane along a transversal transect through the mooring location from the simulation initialized with temperature profiles $T_r$ (**c**, **d**) and $T_{ct}$ (**e**, **f**). Green lines: zero-isotachs marking the nodal lines. Distance is measured from the northern shore. Note the stronger lower hypolimnion currents and deeper nodal lines for the case of $T_r$. Data are bandpass-filtered (V2: 14.05–14.55 h, and V3: 14.75–15.25 h). Idealized simulation for 2022.

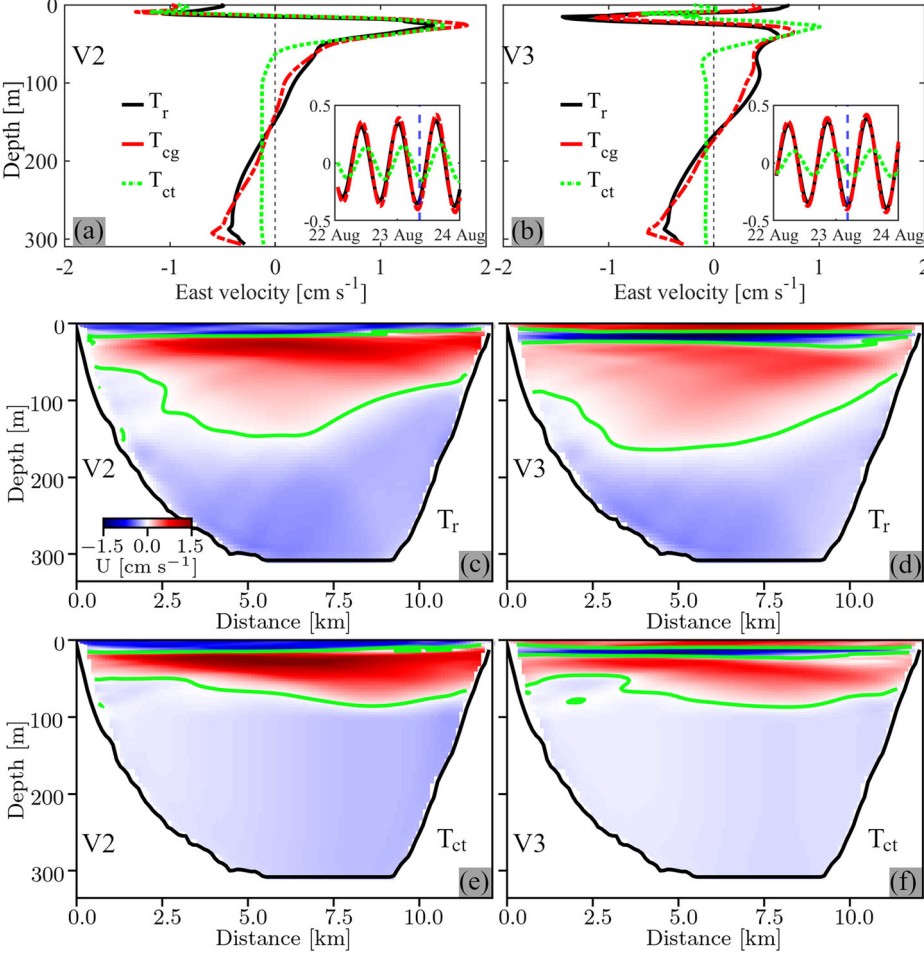

## Conclusions and implications

Combining field observations and 3D numerical modeling, we discovered vertical mode-two (V2) and vertical mode-three (V3) Poincaré waves in large, 309-meter deep Lake Geneva and demonstrated how they dominate the dynamics in the deepest layers during summer stratification, generating currents of up to 4 cm s$^{-1}$ at ~300-m depth. Furthermore, we revealed that weak hypolimnetic stratification ($N^2 \approx 10^{-6}\ s^{-2}$), characteristic of deep lakes, fundamentally changes the vertical structure of higher vertical-mode seiches compared to a hypolimnion without stratification. Although relatively weak, hypolimnetic stratification is important because it induces shear in the middle of the hypolimnion and strengthens near-bottom currents. This has significant implications for hypolimnetic mixing, sediment-water exchange and biogeochemical lake system development, since in deep lakes, the hypolimnion below 100-m depth may contain a large volume and sediment surface; in Lake Geneva this amounts to ~48% of the total volume and ~60% of the sediment surface. Our analysis has made evident that classical concepts such as those based on multiple constant-temperature (density) layers cannot correctly characterize higher vertical-mode seiche dynamics in deep lakes with weak hypolimnetic stratification. Although mixing and transport processes directly driven by wind barely reach below 100-m depth in large deep lakes during summer stratification, internal seiches generated by the same wind, in particular V2 and V3 Poincaré waves, can produce significant deepwater movements even down to the deepest layers, as we have demonstrated here.

The findings of the present study will become even more pertinent in the future since persistent global warming will further extend the stratification period[58,59] and thus the period when Poincaré waves can be generated. On the other hand, a longer stratification period will shorten the winter period during which convective cooling can contribute to deepwater renewal. This shift will lead to a different annual deepwater cycle in the future. Therefore, improving the understanding of deepwater dynamics in deep lakes is becoming increasingly important, in particular, since the role of higher vertical-mode (Poincaré) seiches in generating strong deepwater currents apparently has not been investigated in any deep (>100-m depth) lake prior to this study. The higher mode Poincaré waves that we detected can also be expected to occur in similar large, deep lakes.

## Methods
### Study site and observations

Lake Geneva (local name: *Lac Léman*; Switzerland/France) is a ~14-km wide, ~73-km long and ~309-m deep oligomictic lake. The mountainous topography channels two large-scale winds from the southwest (*Vent*) and northeast (*Bise*) (Fig. 1a), which drive most of the lake's circulation[60–67] and induce V1 Poincaré and Kelvin waves[2,23].

A mooring was deployed in the lake's center at ~305-m depth (Fig. 1a) during the summer from 1 June to 1 September in 2021 and 2022. In 2021, one downward-looking Acoustic Doppler Current Profiler (ADCP) at ~280-m depth measured currents in the lowest ~20 m of the water column. Temperatures were recorded in the lowest 50 m. In 2022, two upward-looking ADCPs at ~290 and ~55-m depth provided full-column measurements (mooring details in Supplementary Table 1). Full-depth temperature profiles were taken twice a month at monitoring station SHL2 (for location, see Fig. 1a) by CIPEL[51].

### Hydrodynamic model

The 3D numerical model (MITgcm[68]) employed here has been validated for Lake Geneva[42]. Following Reiss et al.[66,67], the model employed a uniform horizontal Cartesian grid with a resolution of 113 m and 100 size-varying

**Fig. 8 | Vertical structure of V1, V2, and V3 modes.** Left column (**a, c, e**): Bandpass-filtered (V1: 9.05–9.55 h, V2: 14.05–14.55 h, and V3: 14.75–15.25 h) east velocity profiles at the mooring location for different initial temperature profiles (see legend and Fig. 4e), normalized to the maximum velocity in the thermocline layer. Right column (**b, d, f**): Corresponding theoretical horizontal velocity eigenfunctions for different initial temperature profiles, normalized to the maximum value in the thermocline layer. The eigenfunctions were obtained by solving the Taylor-Goldstein equation without background current (for details, see Methods section).

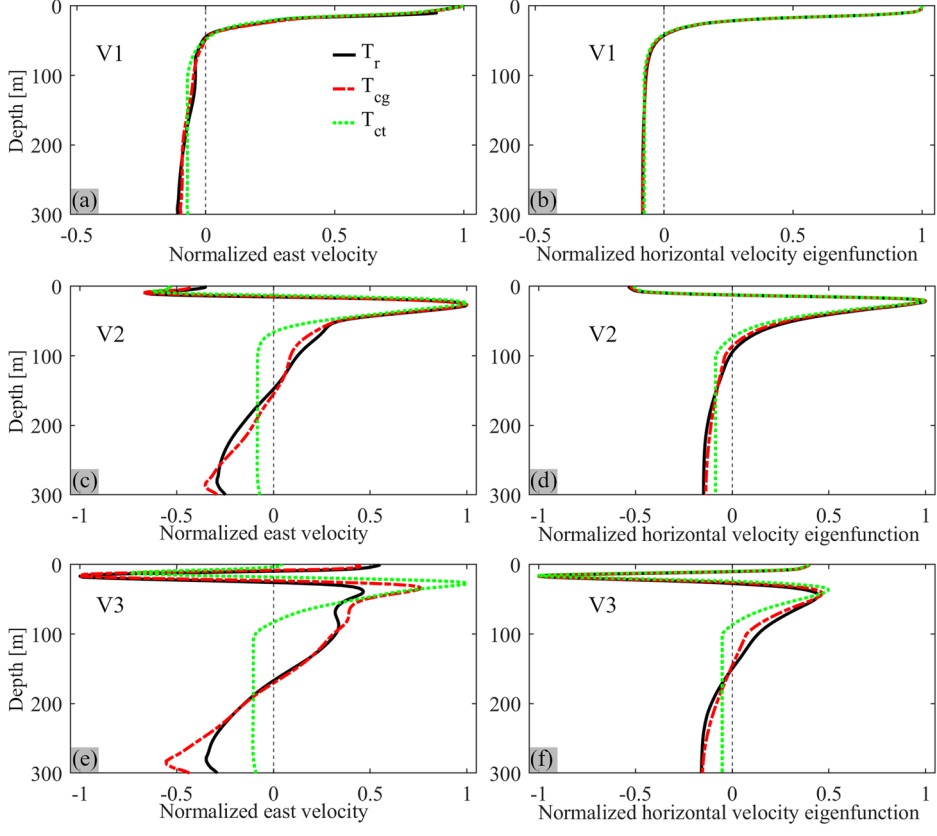

z-layers (30 cm at the surface and 4.8 m at the lake's deepest point). Density as a function of temperature and pressure was computed with a 25-term equation-of-state[69] with salinity kept constant at 0.03 psu; salinity plays a minor role in determining water density in Lake Geneva. The wind stress vector, $\tau$, was computed as $\tau = C_D \rho_a \Delta U_{10} |\Delta U_{10}|$, with $\rho_a$ being the density of air, $\Delta U_{10} = U_{10} - U_0$ being the difference between the wind velocity vector at 10 m above the lake surface, $U_{10}$, and the surface current vector, $U_0$, and $C_D$ being the wind speed-dependent drag coefficient as described by Large and Yeager[70].

Two types of simulations were performed: (i) Two realistic simulations in 2021 and 2022, with several months of spin-up and realistic external forcing, and (ii) four different idealized simulations that were initialized from rest with horizontally homogeneous temperatures, forced at the beginning with a 4-h wind impulse and run for ~20 d.

The realistic simulations were initialized from rest on 20 August 2020 at 11:00 CET (2021 model run) and 15 September 2021 at 12:00 (2022 model run) with horizontally homogeneous temperatures derived from the temperature profiles taken at CIPEL monitoring station SHL2 (for location, see Fig. 1a) and forced with hourly output from the COSMO-1 numerical weather model (resolution 1.1 km) provided by MeteoSwiss.

The idealized simulations were initialized from rest with horizontally homogeneous temperatures derived from the 2022 realistic simulation by time-averaging the modeled temperature profile at station SHL2 (for location, see Fig. 1a) from 17 to 19 August 2022 (referred to as profile $T_r$; see Figs. 1j and 4e). To study how hypolimnetic stratification impacts the structure of vertical mode-two (V2) and vertical mode-three (V3) Poincaré waves, two additional idealized simulations were run with the initial temperature profile above 100-m depth identical to profile $T_r$. However, below that depth, in one simulation, the temperature gradient was kept constant (profile $T_{cg}$) and in the other one, temperatures were kept constant (profile $T_{ct}$) (Fig. 4e).

To efficiently excite near-inertial Poincaré waves while keeping other basin-scale motions comparably low, the idealized simulations were forced only at the beginning with a realistic, strong (~10 m s⁻¹) 4-h wind impulse

derived from the COSMO-1 wind field during either *Bise* (from the northeast; three simulations, initialized with profiles $T_r$, $T_{cg}$, and $T_{ct}$) or *Vent* (from the southwest; one simulation, initialized with profile $T_r$) wind conditions (Supplementary Fig. 10; also see Fig. 1a).

The realistic simulations served to validate the model. The idealized simulations, on the other hand, allowed exploring details of the internal waves, including the effect of different hypolimnetic stratifications and wind forcing.

## Analysis methods

Multitaper rotary current spectra and wavelet transforms were computed using the MATLAB package of Lilly[71]. Furthermore, to analyze internal wave characteristics, such as the vertical and horizontal wave structures, model results and field observations were bandpass-filtered with a phase-preserving Butterworth filter centered around the corresponding wave period/frequency (V1 Poincaré wave: ~9.3 h, V2 Poincaré wave: ~14.3 h, and V3 Poincaré wave: ~15.0 h).

## Vertical mode structure

The theoretical shape of the first three vertical modes for a given initial temperature profile was estimated by solving the Taylor-Goldstein equation[53]

$$\frac{\partial^2 \phi(z)}{\partial z^2} + \left( \frac{N^2(z)}{(\bar{U}(z) - c)^2} - \frac{\partial^2 \bar{U}/\partial z^2}{\bar{U}(z) - c} - k^2 \right)\phi = 0 \quad (1)$$

where $\phi(z)$ denotes the vertical structure or streamfunction, $\bar{U}$ the background horizontal current, $c$ the phase speed, and $k$ the horizontal wavenumber. $N^2(z) = -g\rho_0^{-1}(\partial\rho/\partial z)$ is the squared buoyancy frequency and $\rho_0$ a reference density. At the lake bottom and surface, $\phi(z = 0) = \phi(z = -D) = 0$.

Equation (1) was solved for the three different initial temperature (stratification) profiles (Fig. 4e and Supplementary Fig. 3) using the

MATLAB code provided by Smyth[72], with $\bar{U} = 0$ (no background current). The horizontal wavenumber, $k_m$, corresponding to the horizontal mode $m$, was set to $k_m = m\pi/L$, where $L$ is the basin length[17]. As previously reported[12], the solutions of Eq. (1) are not sensitive to the choice of the horizontal order or to using basin width instead of basin length as a characteristic dimension. Note that Eq. (1) does not consider rotational effects, which is justified[25] because $f \sim \omega \ll N_{\max}$, where $f$ is the latitude-dependent Coriolis-parameter and $\omega$ is the angular wave frequency, both O $(10^{-4}\ s^{-1})$.

## Data availability

The in situ data and model results supporting the findings of this study are available online at https://doi.org/10.5281/zenodo.13143847.

## Code availability

The main MITgcm model configuration files are available online at https://doi.org/10.5281/zenodo.13144189.

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

## Acknowledgements
This work was supported by the Swiss National Science Foundation (Grant numbers 159422 and 217960) and the Bois Chamblard Foundation. We thank Htet Kyi Wynn and Valentin Kindschi for fieldwork assistance.

## Author contributions
All authors contributed substantially to the study's conception. R.S. Reiss was responsible for data acquisition, modeling, data analysis, and drafting/revising the manuscript. U. Lemmin was involved in the design of the field campaign, data interpretation, discussion, and drafting/revising the manuscript. C. Monin contributed to data analysis and discussion. D.A. Barry contributed to data interpretation, discussion, and revising the manuscript.

## Competing interests
The authors declare no competing interests.
