## [Transparent Peer Review file · Communications Earth & Environment]

Strong bottom currents in large, deep Lake Geneva generated by higher vertical-mode Poincaré waves

Corresponding Author: Dr Rafael Reiss

Version 0:

Decision Letter:

Dear Dr Reiss,

Your manuscript titled "Newly discovered vertical mode-two Poincaré waves drive strong bottom currents in large, deep Lake Geneva" has now been seen by 2 reviewers, and we include their comments at the end of this message. They find your work of interest, but some important points are raised. We are interested in the possibility of publishing your study in Communications Earth & Environment, but would like to consider your responses to these concerns and assess a revised manuscript before we make a final decision on publication.

We therefore invite you to revise and resubmit your manuscript, along with a point-by-point response that takes into account the points raised. Please highlight all changes in the manuscript text file. Along with addressing the reviewers comments, to be considered for publication, please take note of the following editorial thresholds:

- Provide more compelling evidence for the V2 Poincare wave structure.
- A more thorough review of existing literature

Please use the following link to submit your revised manuscript, point-by-point response to the referees' comments (which should be in a separate document to any cover letter), a tracked-changes version of the manuscript (as a PDF file) and the completed checklist:

Link Redacted

We hope to receive your revised paper within six weeks; please let us know if you aren't able to submit it within this time so that we can discuss how best to proceed. If we don't hear from you, and the revision process takes significantly longer, we may close your file. In this event, we will still be happy to reconsider your paper at a later date, as long as nothing similar has been accepted for publication at Communications Earth & Environment or published elsewhere in the meantime.

Please do not hesitate to contact us if you have any questions or would like to discuss these revisions further. We look forward to seeing the revised manuscript and thank you for the opportunity to review your work.

Best regards,

Jennifer Veitch, PhD
Editorial Board Member
Communications Earth & Environment
orcid.org/0000-0003-2544-1243

Clare Davis, PhD
Senior Editor
Communications earth & Environment

EDITORIAL POLICIES AND FORMATTING

Editorial Policy: [Policy requirements](https://www.nature.com/documents/nr-editorial-policy-checklist.pdf) (Download the link to your computer as a PDF.)

Furthermore, please align your manuscript with our format requirements, which are summarized on the following checklist: [Communications Earth & Environment formatting checklist](https://www.nature.com/documents/commsj-phys-style-formatting-checklist-article.pdf)

and also in our style and formatting guide [Communications Earth & Environment formatting guide](https://www.nature.com/documents/commsj-phys-style-formatting-guide-accept.pdf) .

***** DATA:** Communications Earth & Environment endorses the principles of the Enabling FAIR data project (<http://www.copdess.org/enabling-fair-data-project/>). We ask authors to make the data that support their conclusions available in permanent, publically accessible data repositories. (Please contact the editor if you are unable to make your data available).

All Communications Earth & Environment manuscripts must include a section titled "Data Availability" at the end of the Methods section or main text (if no Methods). More information on this policy, is available at <http://www.nature.com/authors/policies/data/data-availability-statements-data-citations.pdf>.

If a community resource is unavailable, data can be submitted to generalist repositories such as [figshare](https://figshare.com/) or [Dryad Digital Repository](http://datadryad.org/). Please provide a unique identifier for the data (for example a DOI or a permanent URL) in the data availability statement, if possible. If the repository does not provide identifiers, we encourage authors to supply the search terms that will return the data. For data that have been obtained from publically available sources, please provide a URL and the specific data product name in the data availability statement. Data with a DOI should be further cited in the methods reference section.

REVIEWER COMMENTS:

Reviewer #1 (Remarks to the Author):

The paper by Reiss et al. provides a comprehensive and intriguing description of a newly discovered V2 Poincaré wave in Lake Geneva. I enjoyed reading the paper, which is perfectly written and provides all details necessary for understanding and being convinced by the authors. We can safely say that V2 Poincaré waves develop in Lake Geneva and are responsible of near-bottom currents, enhanced by weak and often neglected stratification. I expect that further studies in similar lakes will confirm these findings. Hence I recommend the publication of this paper after a few very minor revisions.

1) Authors claim that this second-mode Poincaré wave has received little attention in the literature and report many studies focusing on the more commonly observed first mode. I believe the bibliography is sufficient, I would recommend including Hodeges et al. (<https://doi.org/10.4319/lo.2000.45.7.1603>) who modeled basin-scale internal waves in Lake Kinneret (and compared with field data), including V2 Poincaré.

2) Line 42: The authors know very well that Coriolis force can modify circulation not only in large lakes, as Ekman-type currents can develop in small lakes or in narrow region of large lakes (Reiss et al. 2022, Amadori et al. 2020). I understand they are referring here to the cases where Coriolis force can modify the internal wave field, which indeed applies to large lakes where $S < 1$. I suggest modifying "circulation" with "internal wave field".

3) Lines 53-62; 80-82. I agree there are not many works out there investigating deep hypolimnion dynamics with such a high resolution data, however the authors are missing van Haren et al. 2021 (<https://doi.org/10.4081/jlimnol.2020.1983>) and van

Haren and Dijkstra 2021 (<https://doi.org/10.1007/s10652-020-09774-2>) who quantified internal wave motions and overturning in the deep lake Garda.

4) Figure 4: authors first describe panels efgh and then abcd. I suggest inverting the plots.

5) Line 178-179: authors state: "As in the measured velocity profiles, the modeled V2 current velocities in the deep hypolimnion noticeably increase towards the bed and are significantly larger than the V1 current velocities". I don't see such a "significant larger" velocity in V2 from fig. 4(d vs h). Can the authors report a number (e.g. velocity at the bottom) to support the statement?

6) Line 190-191: authors state "contributions of V2-mode bottom currents are one order of magnitude greater than V1-mode bottom currents (Figure 4e)". I guess they refer to one order of magnitude in the spectrum. Please clarify, as PSD has not the same units as velocity.

7) Lines 215-216: I actually see three layers in fig. 5f. Maybe improving the visibility of the surface layer would help.

8) Line 252 and 262: the authors refer to a profile but refer to figure 2c, where the spectrum is displayed. Is that correct? If yes, please clarify.

9) Figure 5: panels a) and b). Grey lines are not very clear; panels c) to j): arrows on the blue are not visible.

10) Line 293: same as in point 6)

11) Lines 378-383. References to all authors' contributions where this model has been used should be avoided. The processes mentioned (nearshore boundary layer, river inflow dynamics, Ekman-driven coastal upwellings, wind-induced interbasin exchange and so on) have no relation with the topic of this contribution. The papers where the model validation (Cimatoribus et al. 2018) and the adopted setup (Reiss et al. 2022) come from are the only meaningful references here.

12) Supplementary: I am quite confused by the supplementary figure 1. Did I get it right that fig S1 a,b = fig. 3 b,c and fig S1c = fig 1b? If yes, why repeating figures? If not, please clarify what kind of additional information we are supposed to obtain from Supplementary figures.

Reviewer #2 (Remarks to the Author):

Reiss et al provide a thorough analysis of internal Poincare waves in Lake Geneva. Their motivation, that these waves have not been studied, particularly in deep lakes, is sound and the results are novel. It is particularly interesting that they V2 waves drive such strong near-bed currents and the importance of weak stratification on the hypolimnion.

I have two major concerns:

(1) There is significant literature missing.

i) the work on Lake Michigan by Troy's group in particular Choi et al (JGR 2012);

(ii) the work by Mortimer and Csanady on Lake Michigan and Lake Ontario (e.g., Mortimer (2012, L&O), Csanady (1973, JPO);

(iii) while for shallower systems (Lake Kinneret and Lake Biwa), the horizontal plane modal analysis by Shimizu and Imberger (2007, 2008 and 2010, L&O) needs to be cited. They extend the modal analysis in the present work to account for arbitrary bathymetry.

(2) I am not convinced you are seeing a V2 Poincare wave, when Fig 2c clearly shows V3 structure.

Specific comments:

- Line 29: Internal seiches in rotational systems (internal Kelvin and Poincare waves) are progressive, not standing.

- Lines 69-80: Lake Michigan (281 m deep) and Lake Ontario (244 m deep) studies should be mentioned here.

Choi, Jun, et al. "A year of internal Poincaré waves in southern Lake Michigan." *Journal of Geophysical Research: Oceans* 117.C7 (2012).

Ahmed, Sultan, Cary D. Troy, and Nathan Hawley. "Spatial structure of internal Poincaré waves in Lake Michigan." *Environmental Fluid Mechanics* 14 (2014): 1229-1249.

Mortimer, C. H. "Inertial oscillations and related internal beat pulsations and surges in Lakes Michigan and Ontario." *Limnology and oceanography* 51.5 (2006): 1941-1955.

Csanady, G. T. "Transverse internal seiches in large oblong lakes and marginal seas." *Journal of Physical oceanography* 3.4 (1973): 439-447.

Lines 80-82: "Full-depth current and temperature profiles that allow investigating details of higher mode Poincaré wave dynamics in the deepest layers of large deep lakes over a full season have thus far not been reported". This statement is *false*.

See for example from Lake Michigan: Choi, Jun, et al. "A year of internal Poincaré waves in southern Lake Michigan." *Journal of Geophysical Research: Oceans* 117.C7 (2012).

And from Lake Ontario: Boegman, Leon, and Yerubandi R. Rao. "Process oriented modeling of Lake Ontario hydrodynamics." *Proceedings of the 16th International Symposium on Environment Hydraulics*. 2010.

Line 123: Please plot the vertical velocity from the ADCP. This will give the wave function and clearly show the vertical modal structure.

Lines 154-155: While the models cannot be easily disentangled spectrally, a model decomposition is typically used (e.g., papers by Antenucci), not modelling. Although, you could cite Saggio and Imberger (L&O, 1998), who do both.

Lines 175-185: The three-layer current structure described here is characteristic to a V3 wave, not a V2 wave. Two layers along up, one layer going down is V3, not V2. Fig 2c also clearly shows a V3 current signature. I'm not convinced these are simply a superposition of different horizontal mode V2 Poincare waves. I'm not saying you are incorrect, but rather you need to do a better job of convincing me, rather than being speculative.

Line 296: I wonder if your dispersion relation solution predicts this could indeed be a V3 Poincare wave? Does your Taylor-Goldstein solution emit a V3 wave eigenfunction that could be compared to the vertical velocity signal from the ADCP?

Line 338: 'Vertical' should be 'vertical'

Communications Earth & Environment is committed to improving transparency in authorship. As part of our efforts in this direction, we are now requesting that all authors identified as 'corresponding author' create and link their Open Researcher and Contributor Identifier (ORCID) with their account on the Manuscript Tracking System prior to acceptance. ORCID helps the scientific community achieve unambiguous attribution of all scholarly contributions. You can create and link your ORCID from the home page of the Manuscript Tracking System by clicking on 'Modify my Springer Nature account' and following the instructions in the link below. Please also inform all co-authors that they can add their ORCIDs to their accounts and that they must do so prior to acceptance.

Author Rebuttal letter: The author's response to these comments can be found at the end of this file.

Version 1:

Decision Letter:

Dear Dr Reiss,

Your manuscript titled "Newly discovered higher vertical-mode Poincaré waves drive strong bottom currents in large, deep Lake Geneva" has now been seen by the original reviewer 1 and a new reviewer 3, who replaces the original reviewer 2. The reviewers' comments are included at the end of this message. In light of their advice we are delighted to say that we are happy, in principle, to publish a suitably revised version in *Communications Earth & Environment*.

We therefore invite you to revise your paper one last time to address the remaining concerns of our reviewers. At the same time we ask that you edit your manuscript to comply with our format requirements and to maximise the accessibility and therefore the impact of your work.

EDITORIAL REQUESTS:

****Please take care to match our formatting and policy requirements. We will check revised manuscript and return manuscripts that do not comply. Such requests will lead to delays. ****

SUBMISSION INFORMATION:

OPEN ACCESS:

Communications Earth & Environment is a fully open access journal. Articles are made freely accessible on publication. For further information about article processing charges, open access funding, and advice and support from Nature Research, please visit <https://www.nature.com/commsenv/open-access>

Link Redacted

Best regards,

Dr Alireza Bahadori
Associate Editor
Communications Earth & Environment

REVIEWERS' COMMENTS:

Reviewer #1 (Remarks to the Author):

The manuscript has substantially changed from the original submission. The authors addressed all my comments and included a new section describing V3 Poincaré waves in addition to V2.
I have no further comments

Reviewer #3 (Remarks to the Author):

This article presents important observations of higher Poincaré wave modes in a mid-sized lake via ADCP observations, theories, and numerical simulations. I recommend a minor revision.

Lines 93-94: In large lakes, such as the Laurentian Great Lakes, higher modes cannot be spectrally separated as the frequencies of all modes converge to the inertial frequency, rendering band-pass filtering ineffective. However, in medium-sized lakes like Lake Geneva, spectral separation of higher modes is feasible due to a Burger number on the order of 0.1. Thus, while it is true that higher modes have not been reported in large lakes, their observation remains still unlikely. This point needs to be addressed.

Figure 1: The spectral range near the inertial period requires focus. Due to the difficulty in differentiating peaks near the inertial period, I recommend using a semilogx scale (rather than loglog) in the spectrum plot. Please add lines at V1, V2, V3, and inertial frequencies in panels b and c.

Figure 1c: Please convert the wind speed to wind stress.

Figure 5, Panels e and f: What do the horizontal lines represent? What is 'D' in the y-axis label? What are the meanings of the red-dotted and black solid lines? I am unclear about the representation of isotherm displacement in the vertical direction at a single location.

Supplementary Figure 5: Please consider using different names for T1, T2, T3 as they can be confused with Tr and T (period).

Author Rebuttal letter: The author's response to these comments can be found at the end of this file.

Manuscript no. COMMSENV-23-1899**Newly discovered higher vertical-mode Poincaré waves drive strong bottom currents in large, deep Lake Geneva****Rafael Sebastian Reiss, Ulrich Lemmin, Claire Monin, David Andrew Barry**

RESPONSES TO THE REVIEWERS

REVIEWER 1

We thank the reviewer for the constructive comments and suggestions (in blue italics), which we have addressed below (in black type).

Comment 1:

Authors claim that this second-mode Poincaré wave has received little attention in the literature and report many studies focusing on the more commonly observed first mode. I believe the bibliography is sufficient, I would recommend including Hodeges et al. (<https://doi.org/10.4319/lo.2000.45.7.1603>) who modeled basin-scale internal waves in Lake Kinneret (and compared with field data), including V2 Poincaré.

This reference is now cited (line 56).

Note that for the revised version of the manuscript, we carried out further analyses which revealed that V3 Poincaré waves are also present in Lake Geneva. These waves have a period close to the V2 Poincaré period discussed in the original manuscript and are of similar strength. The manuscript was revised accordingly (see highlighted text in the manuscript).

Comment 2:

Line 42: The authors know very well that Coriolis force can modify circulation not only in large lakes, as Ekman-type currents can develop in small lakes or in narrow region of large lakes (Reiss et al. 2022, Amadori et al. 2020). I understand they are referring here to the cases where Coriolis force can modify the internal wave field, which indeed applies to large lakes where $S < 1$. I suggest modifying “circulation” with “internal wave field”.

The text was modified (line 44).

Comment 3:

Lines 53-62; 80-82. I agree there are not many works out there investigating deep hypolimnion dynamics with such a high resolution data, however the authors are missing van Haren et al. 2021 (<https://doi.org/10.4081/jlimnol.2020.1983>) and van Haren and Dijkstra 2021 (<https://doi.org/10.1007/s10652-020-09774-2>) who quantified internal wave motions and overturning in the deep lake Garda.

The two references were added (lines 88-91).

Comment 4:

Figure 4: authors first describe panels e f g h and then a b c d. I suggest inverting the plots.

The panel order was changed. Note that Figure 4 now shows modes V2 and V3, instead of modes V1 and V2.

Comment 5:

Line 178-179: authors state: “As in the measured velocity profiles, the modeled V2 current velocities in the deep hypolimnion noticeably increase towards the bed and are significantly larger than the V1 current velocities”. I don’t see such a “significant larger” velocity in V2 from fig. 4(d vs h). Can the authors report a number (e.g. velocity at the bottom) to support the statement?

This section was entirely revised. The contribution of the different modes is now discussed based on basin-wide integrated kinetic energy and bottom velocities at the mooring location in the new section, *Relative strengths of V1, V2 and V3 Poincaré waves*.

Comment 6:

Line 190-191: authors state “contributions of V2-mode bottom currents are one order of magnitude greater than V1-mode bottom currents (Figure 4e)”. I guess they refer to one order of magnitude in the spectrum. Please clarify, as PSD has not the same units as velocity.

This paragraph was removed. See also our response to Comment 5 above.

Comment 7:

Lines 215-216: I actually see three layers in fig. 5f. Maybe improving the visibility of the surface layer would help.

This section was entirely revised. We now focus on discussing the difference between Poincaré modes V2 and V3 along one transect passing through the mooring location. Figure 5 was modified. Results for additional transects are given in Supplementary Figure 5 and Supplementary Movie 1.

Comment 8:

Line 252 and 262: the authors refer to a profile but refer to figure 2c, where the spectrum is displayed. Is that correct? If yes, please clarify.

This section was entirely revised.

Comment 9:

Figure 5: panels a) and b). Grey lines are not very clear; panels c) to j): arrows on the blue are not visible.

This problem appears to be due to the reduced quality of the pdf version created automatically for the review process. In the original document that we had uploaded, the colors and arrows were clearly distinguishable when zooming into the figure. Figure 5 was modified (see response to Comment 7 above).

Comment 10:

Line 293: same as in point 6)

The text was modified. Note that the strength of the modes is now discussed in the new section, *Relative strengths of V1, V2 and V3 Poincaré waves*. See also our response to Comment 5 above. We believe that Figure 1g clearly displays the difference in strength between V1 currents and V2/V3 currents.

Comment 11:

Lines 378-383. References to all authors' contributions where this model has been used should be avoided. The processes mentioned (nearshore boundary layer, river inflow dynamics, Ekman-driven coastal upwellings, wind-induced interbasin exchange and so on) have no relation with the topic of this contribution. The papers where the model validation (Cimatoribus et al. 2018) and the adopted setup (Reiss et al. 2022) come from are the only meaningful references here.

The text was modified (lines 443-444).

Comment 12:

Supplementary: I am quite confused by the supplementary figure 1. Did I get it right that fig S1 a,b = fig. 3 b,c and fig S1c = fig 1b? If yes, why repeating figures? If not, please clarify what kind of additional information we are supposed to obtain from Supplementary figures.

Supplementary Figure 1a, b and Figure 3b, c show the same data. These two panels were repeated in Supplementary Figure 1 (mentioned in the caption) to mark in the time series plots the two periods for which progressive vector diagrams are shown in Supplementary Figure 1d, e.

Supplementary Figure 1c and Figure 1b are similar but do not show all the same curves. Figure 1b shows the modeled and observed clockwise rotary current spectra and the observed counterclockwise rotary current spectrum. Supplementary Figure 1c shows the observed temperature spectrum and the observed clockwise rotary current spectrum. The latter curve was repeated in the Supplementary Figure 1c for easier comparison with the temperature spectrum. This was clarified in the figure caption.

REVIEWER 2

We thank the reviewer for the constructive comments and suggestions (in blue italics), which we have addressed below (in black type).

Major comment 1:

There is significant literature missing.

i) the work on Lake Michigan by Troy's group in particular Choi et al (JGR 2012);

(ii) the work by Mortimer and Csanady on Lake Michigan and Lake Ontario (e.g., Mortimer (2012, L&O), Csanady (1973, JPO));

(iii) while for shallower systems (Lake Kinneret and Lake Biwa), the horizontal plane modal analysis by Shimizu and Imberger (2007, 2008 and 2010, L&O) needs to be cited. They extend the modal analysis in the present work to account for arbitrary bathymetry.

We now cite four studies by Troy's group in Lake Michigan, including Choi et al. (2012) (lines 74-79), Csanady (1973) (lines 182-184 and 282-284), Mortimer (2006) (lines 181-182, 247-250, 279-281 and 329-330), and three studies by Shimizu et al. (lines 182-184).

Although complementary, these studies do not deal with the focus of this paper, i.e., the significant effect that V2 and V3 Poincaré waves can have on deep hypolimnion hydrodynamics.

Major comment 2:

I am not convinced you are seeing a V2 Poincare wave, when Fig 2c clearly shows V3 structure.

We carried out additional detailed analyses which indeed revealed that both V2 and V3 Poincaré waves with very similar periods are present in the observations and model results. The relevant passages and figures (particularly Figures 2 and 4-8 and Supplementary Figures 3-5, 7 and 8) were revised throughout the manuscript. For further details, see our responses to Specific comments 4, 6 and 7 below.

Specific comment 1:

Line 29: Internal seiches in rotational systems (internal Kelvin and Poincare waves) are progressive, not standing.

It is correct that these waves are not standing waves in the strictest sense. However, since Kelvin/Poincaré waves are formed by the superposition of two oppositely-propagating Kelvin/Poincaré waves, the term "standing wave" is often used in the literature, in analogy to a true standing seiche without rotation.

We believe the term "quasi-standing" is suitable here to distinguish Kelvin/Poincaré waves from truly propagating waves (for a discussion, see, e.g., the textbook by Hutter et al. (2011)). This was clarified in lines 51-54.

Specific comment 2:

Lines 69-80: Lake Michigan (281 m deep) and Lake Ontario (244 m deep) studies should be mentioned here.

Choi, Jun, et al. "A year of internal Poincaré waves in southern Lake Michigan." Journal of Geophysical Research: Oceans 117.C7 (2012).

Ahmed, Sultan, Cary D. Troy, and Nathan Hawley. "Spatial structure of internal Poincaré waves in Lake Michigan." *Environmental Fluid Mechanics* 14 (2014): 1229-1249.

Mortimer, C. H. "Inertial oscillations and related internal beat pulsations and surges in Lakes Michigan and Ontario." *Limnology and oceanography* 51.5 (2006): 1941-1955.

Csanady, G. T. "Transverse internal seiches in large oblong lakes and marginal seas." *Journal of Physical oceanography* 3.4 (1973): 439-447..

These references are now cited: Choi et al. (2012) and Ahmed et al. (2014) (lines 74-79), Csanady (1973) (lines 182-184 and 282-284), and Mortimer (2006) (lines 181-182, 247-250, 279-281 and 329-330) who was already cited in the original manuscript (lines 49, 233-235).

Although these four studies have been conducted in deep lakes (244 and 281 m), none has focused on the deepest layers (maximum depth of the reported observations is ~150 m). See also our responses to Major comment 1 and Specific comment 3. See also text lines 74-79.

Specific comment 3:

*Lines 80-82: "Full-depth current and temperature profiles that allow investigating details of higher mode Poincaré wave dynamics in the deepest layers of large deep lakes over a full season have thus far not been reported". This statement is *false*.*

*See for example from Lake Michigan: Choi, Jun, et al. "A year of internal Poincaré waves in southern Lake Michigan." *Journal of Geophysical Research: Oceans* 117.C7 (2012).*

*And from Lake Ontario: Boegman, Leon, and Yerubandi R. Rao. "Process oriented modeling of Lake Ontario hydrodynamics." *Proceedings of the 16th International Symposium on Environment Hydraulics*. 2010.*

We believe that this statement is correct because we have not found any such observations for the deepest layers of a comparably deep lake.

Choi et al. (2012) report full water column temperature measurements, this however, only at a ~150-m deep location (max. depth of Lake Michigan is 281 m). Furthermore, their current measurements only consisted of four single-point current meters at 12, 22 117 and 154-m depth. Thus, they could not resolve higher vertical modes, as the authors note in that paper.

Boegman and Rao (2010) reported temperature and current measurements from a ~90-m deep location (max. depth of Lake Ontario is 244 m).

Specific comment 4:

Line 123: Please plot the vertical velocity from the ADCP. This will give the wave function and clearly show the vertical modal structure.

The measured bandpass-filtered vertical velocities are very low and noisy, and do not reveal any new information. This can be partly explained by the low particle concentration at mid-depth (lines 179-181). More important, the mooring location is situated near an amphidromic point of the V2 and V3 Poincaré waves (see new Figure 5a, b), where vertical velocities and isotherm displacements are expected to vanish (see new Supplementary Figures 3 and 4).

Therefore, instead, we now plot the bandpass-filtered vertical isotherm displacement for modes V2 and V3 at a ~200-m deep location north of the mooring (new Figure 5e, f).

Furthermore, the theoretical vertical velocity eigenfunctions of modes V2 and V3, obtained by solving the Taylor-Goldstein equation for the realistic temperature profile, T_r , are given in new Supplementary Figure 8. All these curves confirm that the ~14-15 h oscillations observed in Lake Geneva are due to V2 and V3 Poincaré waves.

Note that the theoretical and modeled profiles of the horizontal velocities for modes V1-V3 are compared in new Figure 8. Good agreement is found, confirming the V2 and V3 modal structures.

See also our responses to Specific comments 6 and 7 below.

Specific comment 5:

Lines 154-155: While the models cannot be easily disentangled spectrally, a model decomposition is typically used (e.g., papers by Antenucci), not modelling. Although, you could cite Saggio and Imberger (L&O, 1998), who do both.

We now cite five studies for different modal decomposition methods, including Saggio and Imberger (1998) (lines 182-184).

Specific comment 6:

Lines 175-185: The three-layer current structure described here is characteristic to a V3 wave, not a V2 wave. Two layers along up, one layer going down is V3, not V2. Fig 2c also clearly shows a V3 current signature. I'm not convinced these are simply a superposition of different horizontal mode V2 Poincaré waves. I'm not saying you are incorrect, but rather you need to do a better job of convincing me, rather than being speculative.

There appears to have been some confusion regarding the discussed velocity components. The text passage referred to by the reviewer and corresponding Figure 4 refer to horizontal currents, not vertical ones. With respect to the horizontal currents, a three-layer current structure is characteristic of a V2 wave. To avoid confusion, the distinction between vertical and horizontal currents is made more explicit in the revised manuscript (e.g., lines 211, 215, 223, and 370-372).

However, as pointed out by the reviewer, the measured velocity profiles (original Figure 2c) were less clear than the model results. Further analysis showed that, in fact, both the vertical modes V2 and V3 are present in the observations and model results. Due to the similar wave periods of these two modes (less than 1 h apart), they could not be separated in the originally chosen filter passbands of 2-h width. The superposition of these two modes resulted in unexpected phase behavior between adjacent layers. With narrower passbands of 0.5-h width, modes V2 and V3 can clearly be distinguished in the model results (see, e.g., new Figures 4d, h, 5c-f and 6c). The presence of both modes was also confirmed by the measured current profiles on 28 June 2022 and 1 July 2022 (new Figure 2a, b).

Furthermore, good agreement is found between the narrow bandpass-filtered model results and the theoretical wave periods (revised Supplementary Text 1) and vertical structures (new Figure 8 and Supplementary Figure 8) of the modes V2 and V3 obtained by solving the Taylor-Goldstein equation. All these new results confirm that V2 and V3 Poincaré waves caused the dominant ~14-15 h signal observed and modeled in the lake's deepest layers. No signals of vertical Poincaré modes higher than V3 were found.

The manuscript was entirely revised to discuss and compare the newly discovered vertical Poincaré modes V2 and V3.

Note that due to the similar strength and wave periods of vertical modes V2 and V3, their superposition caused a so-called beat pulsation, a phenomenon known from acoustics. This is discussed in the new section, *Beat pulsation by superposition of V2 and V3 Poincaré waves*.

See also our responses to Major comment 2 and Specific comments 4 and 7.

Specific comment 7:

Line 296: I wonder if your dispersion relation solution predicts this could indeed be a V3 Poincaré wave? Does your Taylor-Goldstein solution emit a V3 wave eigenfunction that could be compared to the vertical velocity signal from the ADCP?

The dispersion relation from Antenucci and Imberger (2001) can be applied to any vertical mode order by considering the corresponding non-rotational phase speed in the Burger number (details in Supplementary Text 1). However, the three-layer model employed in the original version of this manuscript to obtain the V2 phase speed did not allow for vertical modes higher than the second order. To overcome this limitation, we now use the non-rotational phase speed obtained by solving the Taylor-Goldstein equation, which allows consideration of higher vertical modes. Furthermore, the new approach is more objective than the three-layer model, because the solutions for the phase speeds are based on the full-depth stratification profile rather than a simplified three-layer profile. Supplementary Text 1 was modified.

With this new approach, we obtain a V2 Poincaré period of 14.7 h (before 14.5 h) and a V3 Poincaré period of 15.7 h for summer 2022 (details in Supplementary Text 1). The theoretical V2 and V3 eigenfunctions obtained by solving the Taylor-Goldstein equation are shown in new Figure 8d, f (horizontal velocities) and in new Supplementary Figure 8 (vertical velocities).

As detailed in the Specific comment 4 above, the measured vertical velocities could not be used. However, further analysis with narrower passbands (0.5-h instead of 2-h width) confirmed the presence of vertical Poincaré modes V2 and V3, as discussed in the sections, *V1, V2 and V3 Poincaré wave features revealed by idealized simulations* and *Horizontal and vertical structure of V2 and V3 Poincaré waves*.

See also our responses to Major comment 2 and Specific comments 4 and 6 above.

Specific comment 8:

Line 338: ‘Vertical’ should be ‘vertical’.

This was done.

Manuscript no. COMMSENV-23-1899A

Strong bottom currents in large, deep Lake Geneva generated by higher vertical-mode Poincaré waves

Rafael Sebastian Reiss, Ulrich Lemmin, Claire Monin, David Andrew Barry

RESPONSES TO THE REVIEWER

REVIEWER 3

We thank the reviewer for the constructive comments and suggestions (in blue italics), which we have addressed below (in black type).

Comment 1:

Lines 93-94: In large lakes, such as the Laurentian Great Lakes, higher modes cannot be spectrally separated as the frequencies of all modes converge to the inertial frequency, rendering band-pass filtering ineffective. However, in medium-sized lakes like Lake Geneva, spectral separation of higher modes is feasible due to a Burger number on the order of 0.1. Thus, while it is true that higher modes have not been reported in large lakes, their observation remains still unlikely. This point needs to be addressed.

This is now clarified in lines 287-292.

Comment 2:

Figure 1: The spectral range near the inertial period requires focus. Due to the difficulty in differentiating peaks near the inertial period, I recommend using a semilogx scale (rather than loglog) in the spectrum plot. Please add lines at V1, V2, V3, and inertial frequencies in panels b and c.

Due to the small panel sizes, changing from a loglog to a semiology representation (i.e., linear representation on the x-axis) does not add significant details in the near-inertial band unless the lower frequencies are cut off. Likewise, marking both V2 and V3 decreases clarity because the lines are too close to each other. Instead, we added larger semiology plots with a zoom on the near-inertial band where the different Poincaré periods are marked separately in the new Supplementary Figure 1.

Comment 3:

Figure 1c: Please convert the wind speed to wind stress.

We added the wind stress curve in Figure 1c and added a sentence and reference on how wind stress is computed in the numerical model (lines 454-458).

Comment 4:

Figure 5, Panels e and f: What do the horizontal lines represent? What is 'D' in the y-axis label? What are the meanings of the red-dotted and black solid lines? I am unclear

about the representation of isotherm displacement in the vertical direction at a single location.

The horizontal dashed lines in panels e and f indicate the location of the thermocline (see figure caption).

“D” was changed to “Depth” in the figures and captions.

The red-dotted and black solid curves give the vertical isotherm displacements at two times half a wave period apart which are now given in the legends. Vertical isotherm displacements at a single location were computed by comparing the respective temperature profile at a given time with a reference temperature profile, here taken as the model’s initial temperature profile, T_r . This was clarified in the figure caption.

Comment 5:

Supplementary Figure 5: Please consider using different names for T1, T2, T3 as they can be confused with T_r and T (period).

We changed the transect names T1, T2, and T3 to S1, S2, and S3 in Supplementary Figures S5 and S7 and Supplementary Movie 1.